# DS-PROVER: A DYNAMIC SAMPLING BASED APPROACH FOR NEURAL THEOREM PROVING

## ABSTRACT

Theorem proving is a fundamental task in mathematics. With the advent of large language models (LLMs) and interactive theorem provers (ITPs) like Lean, there has been growing interest in integrating LLMs and ITPs to automate theorem proving. In this approach, the LLM generates proof steps (tactics), and the ITP checks the applicability of the tactics at the current goal. The two systems work together to complete the proof. In this paper, we introduce DS-Prover, a novel dynamic sampling method for theorem proving. This method dynamically determines the number of tactics to apply to expand the current goal, taking into account the remaining time compared to the total allocated time for proving a theorem. This makes the proof search process more efficient by adjusting the balance between exploration and exploitation as time passes. We also study the effect of augmenting the training dataset by decomposing simplification and rewrite tactics with multiple premises into tactics with single premises. This gives the model more examples to learn from and helps it to predict the tactics with premises more accurately. We perform our experiments using the Mathlib dataset of the Lean theorem prover and report the performance on two standard datasets, MiniF2F and ProofNet. Our methods achieve significant performance gains on both datasets. We achieved a state-of-the-art performance (Pass@1) of 14.2% on the ProofNet dataset and a performance of 29.8% on MiniF2F, slightly surpassing the best-reported Pass@1 of 29.6% using Lean.

## 1 INTRODUCTION

Theorem proving in mathematics is a formal and systematic approach to establishing the validity of mathematical statements, known as theorems. It involves demonstrating that a given mathematical proposition is true based on a set of axioms and logical rules. The process typically follows a structured and rigorous method, and it can be done either manually or with the assistance of computer programs called interactive theorem provers (ITPs). Some popular ITPs include Coq (Barras et al., 1997), Metamath (Megill & Wheeler, 2019), Isabelle (Nipkow et al., 2002), HOL Light (Harrison, 1996), and Lean (de Moura et al., 2015). ITPs are commonly used for the verification of mathematical proofs and formalization of mathematics.

Automating the generation of proofs of mathematical theorems has been a topic of interest for a long time (Thurston, 1994; De Millo et al., 1979). When using machine learning to generate proofs of theorems, two important features of interactive theorem provers (ITPs) are used: formalized proofs, which are used as the dataset for training the machine learning model, and the ability to verify the correctness of proofs. To automate the generation of mathematical proofs, we combine an ITP with a large language model (LLM). The LLM generates possible proof steps (tactics) for a given goal that can be applied to it. The ITP checks the applicability of the generated tactic and creates new edges with the applicable tactics in the proof search tree, discarding the failed tactics. For our task, we used the Lean interactive theorem prover and ByT5 (Xue et al., 2022) as the tactic generator model. ITPs have already been helping mathematicians with their work, and AI-based proof generation can further help them and accelerate research in mathematics.

Although transformer-based language models (Vaswani et al., 2017) are very good at learning the pattern of tactic-goal pairs, we currently have limited formalized proofs in ITP libraries that can be used as a dataset for training the language model. To overcome the issue of limited data for training

the model, various new approaches have been explored to improve the performance of automated theorem provers, such as reinforcement learning (Polu & Sutskever, 2020; Wu et al., 2021a), expert iteration (Polu et al., 2022), online training (Lample et al., 2022) and retrieval augmentation (Yang et al., 2023). Although these methods improve the performance of the prover to some extent, one of the issues with these provers is that they require another model along with a tactic generator model to guide/help the proof search. This makes them computationally expensive to both train and use.

Here, we propose DS-Prover, a new dynamic sampling method for theorem proving. This method dynamically determines the number of tactics to apply to expand the current goal, taking into account the remaining time compared to the total allocated time for proving a theorem. This makes the proof search process more efficient by adjusting the balance between exploration and exploitation as time passes. As an ML model learns to predict the tactics based on the tactic-goal pair it was trained on, we study the effect of augmenting the training dataset by increasing the number of tactic-goal pairs in the training dataset for the model by breaking down the tactics like `rw` (rewrite) and `simp` (simplify) with multiple premises into tactics with single premises. We believe that this will give the model more examples to learn from and help it to predict the tactics more accurately. Further, we optimized the LeanDojo (Yang et al., 2023), which is a tool we used to interact with Lean programmatically. In addition, we introduce a publicly accessible theorem prover website, allowing users to submit the formal statement of their mathematical theorem in Lean. The prover will then endeavor to prove it within a specified time limit.

We conducted our experiments using the Lean formal environment and its formalized theorems in the Mathlib repository as the training dataset. We evaluated our methods on two standard datasets, MiniF2F and ProofNet, and achieved significant performance gains on both. We achieved a state-of-the-art performance (Pass@1) of 14.2% on the ProofNet dataset and a performance of 29.8% on MiniF2F, slightly surpassing the best-reported Pass@1 of 29.6% (Polu et al., 2022) using Lean.

## 2 RELATED WORKS

With the increasing number of formalized theorems in the libraries of interactive theorem provers (ITPs), various recent attempts have been made to automate theorem proving using machine learning. Before the advent of the Large Language Model (LLM), the architecture of machine learning agents used for interacting with the ITPs was based on graph neural networks, as evidenced by studies such as (Yang & Deng, 2019; Paliwal et al., 2020), which could explicitly encode the syntax of formal expressions. Additionally, deep learning-based approaches (Irving et al., 2016; Whalen, 2016; Bansal et al., 2019) were employed for this purpose.

After the success of the attention-based (Vaswani et al., 2017) LLM, in (Polu & Sutskever, 2020) utilized a GPT-2 (Radford et al., 2019) model for tactic generation for a given state, employing a best-first search method to guide the proof using the log-probability of tactics. Additionally, they incorporated a state-level value function to further enhance the guidance of the proof search process. Building upon this, PACT (Han et al., 2021) made strides by leveraging both tactic and term-based proofs from Lean's library Mathlib, utilizing a multi-task training scheme to train their model. Conversely, in (Wang & Deng, 2020), the authors addressed the data scarcity issue by implementing a method where they learned to prove theorems through the process of generating them. Meanwhile, in the work by (Wu et al., 2021b), synthetic datasets were generated for deduction, induction, and abduction tasks, recognized as fundamental reasoning primitives (Peirce, 1992). The model was trained on this data to equip it with proficiency in these three reasoning abilities. Additionally, in TacticZero (Wu et al., 2021a), deep reinforcement learning techniques were employed, enabling the learning of theorem proving from scratch.

To enhance the performance of neural theorem provers on datasets that potentially possess distributions different from the training data, the authors in (Polu et al., 2022) employed an expert iteration-based method. This approach intertwines proof search with learning from previous attempts, aiming to improve future performance. Similarly, HyperTree Proof Search (Lample et al., 2022) employed a comparable strategy, learning from previous proof searches through online training. This method facilitates the model's ability to generalize to domains significantly distant from the training distribution.

Given that most of these works rely on a step-level value function to steer the proof search process, DT-Solver (Wang et al., 2023) presents an alternative approach. The authors utilize a dynamic-tree-driven proof search procedure that integrates state confidence and proof-level values, mirroring human reasoning by considering the entire proof trajectory. One of the difficulties of the proof step generating agent is to find the correct premises to be used in a given proof step. To address this issue, some studies (Szegedy et al., 2021) have been conducted, and LeanDojo (Yang et al., 2023) and Wu (2022) have implemented a retrieval-augmented language model and Magnushammer (Mikuła et al., 2023) have used a transformer-based approach to suggest the premises to be used in tactics. Similar approaches for automating theorem proving have also been applied to other interactive theorem provers like Isabelle (Li et al., 2020; Jiang et al., 2021; First et al., 2023; Jiang et al., 2022).

Although most of the methods discussed above improve performance, one issue is that most of these methods use multiple models and multiple training objectives, leading to processes that require higher computing power (in some cases, up to 1000 GPU hours) for both training and proving. To overcome the issue of high computation demand, we instead explore the possibility of improving the performance with a tactic generator model only. To do this, we improved the prover algorithm by introducing DS-Prover, a Dynamic Sampling method, and explored the option of making the tactic generator model better at premise-based tactic prediction by giving it more pairs of goals and tactics with premises. This helps the model to learn the names of the premises and the goals where to use them.

## 3 FORMAL ENVIRONMENT

For our experiment, we chose Lean (de Moura et al., 2015) as the formal environment. Lean is an interactive proof assistant that is designed to assist mathematicians, computer scientists, and engineers in formalizing and proving the correctness of mathematical theorems and complex software systems. Lean uses a powerful dependent type system, which allows types to depend on values and expressions. This feature makes it possible to express precise and intricate mathematical concepts, making it suitable for formalizing advanced mathematical theories.

Lean's metaprogramming capabilities, active community, extensive mathematical library, and robust tactic system make it a compelling choice for our experiment. These features enable us to effectively write custom automation procedures, access a wealth of formalized mathematical knowledge, and utilize a powerful interactive theorem proving environment. Additionally, Lean's active and growing user community provides support for newcomers and contributes to its ongoing development, ensuring that we can utilize the latest advancements and benefit from the collective expertise of the Lean community.

At the time of conducting our experiments, Lean 4 was the most recent version available. However, due to its lack of backward compatibility and the ongoing process of migrating the mathematical library Mathlib from Lean 3 to Lean 4 (Mathlib4), we chose to utilize Lean 3 for our experiments. It's worth noting that our experimental procedures can readily be replicated within the Lean 4 environment when Mathlib4 is fully established.

### 3.1 PROVING IN LEAN

Proving in Lean involves constructing formal proofs to demonstrate the correctness of mathematical statements. Here's an example:

1. Environment Setup: In this step, we set up the mathematical environment in Lean. This involves declaring variables and stating assumptions if necessary.

   ```
   variables x y z w : nat
   ```

   Here, we declare four natural number variables: `x`, `y`, `z`, and `w` which will be used in the proof.

2. Formulate the Goal: Define the theorem or proposition to be proved.

   ```
   theorem thm1 (h1 : x = y) (h2 : y = z) (h3 : z = w) : x = w :=
   ```

This line defines a theorem statement named 'thm1'. We state our goal, which is to prove that `x = w`. Additionally, we introduce assumptions or hypotheses (`h1 :   x = y`), (`h2 :   y = z`), and (`h3 :   z = w`).

3. Constructing the proof: involves a systematic progression through a series of steps, applying various tactics to achieve the goal. The theorem statement serves as the initial goal, and the application of valid tactics generates one or more subgoals. The proof is ultimately obtained by continually applying new tactics to these subgoals until no further goals remain. In the example presented here, we employ two primary tactics, namely the `rw` (rewrite) and `assumption` tactics, to establish a chain of equalities that underpin the proof.

```
begin
  rw [h1, h2],
  assumption
end
```

- `begin`: This keyword marks the beginning of the proof. It's followed by a sequence of tactics that will be applied in order to achieve the goal.
- `rw [h1, h2]`: We use the `rw` (rewrite) tactic to simplify the goal by applying rewriting rules based on the assumptions `h1` and `h2`. This step simplifies the goal by substituting `x` with `y` and `y` with `z`.
- `assumption`: The `assumption` tactic looks through the assumptions in context of the current goal, and if there is one matching the conclusion, it applies it. In this case, it concludes the proof by using the third assumption `h3` to establish that `z` is indeed equal to `w`.
- `end`: This keyword marks the end of the proof.

## 3.2 DATASET

For our experiment, we used Lean's mathematical library Mathlib [1] with commit `d11f435d4e34a6cea0a1797d6b625b0c170be845` (Date: July 26, 2023) as our dataset which has a total of 98,508 theorems. We created the train, validation, and test datasets randomly with 96,508, 1,000, and 1,000 theorems respectively. To train the tactic generator model we used LeanDojo (Yang et al., 2023) to generate the goal tactic pair for each step in theorems' proof.

### 3.2.1 ORIGINAL DATA

For the training dataset, we had a total of 212,840 goal tactic pairs. Below we demonstrate the goal tactic pairs for the proof of theorem 'thm1' discussed above.

```
[{"goal": x y z w: nat
         h1: x = y
         h2: y = z
         h3: z = w
         |- x = w
  "tactic": "rw [h1, h2]"},
 {"goal": x y z w: nat
         h1: x = y
         h2: y = z
         h3: z = w
         |- z = w
  "tactic": "assumptions"}]
```

### 3.2.2 AUGMENTED DATA

To further increase the number of tactic-goal pairs, we decomposed tactics with multiple premises, such as `rw` $[p_1, ..., p_n]$ and `simp` $[p_1, ..., p_n]$, into tactics with single premises. This means we transformed them into sequences like `rw` $[p_1]$; ...; `rw` $[p_n]$ and `simp` $[p_1]$; ...;

---

[1] https://github.com/leanprover-community/mathlib

simp $[p_n]$. This adjustment allowed us to generate a greater variety of examples illustrating the usage of each individual premise. This approach aids the model in learning the appropriate pairings of premises with specific goals and helps it remember the names of these premises. Table 3 in the Appendix A.1 summarizes all of these modifications.

```
[{"goal": x y z w: nat
         h1: x = y
         h2: y = z
         h3: z = w
         |- x = w
  "tactic": "rw [h1]"},
 {"goal": x y z w: nat
         h1: x = y
         h2: y = z
         h3: z = w
         |- y = w
  "tactic": "rw [h2]"},
 {"goal": x y z w: nat
         h1: x = y
         h2: y = z
         h3: z = w
         |- z = w
  "tactic": "assumptions"}]
```

**Data Collection:**  As the list of premises in the rewrite (rw, erw, rwa, equiv_rw, assoc_rw) tactics are ordered, so to collect this data we modified all the occurrences of the rewrite tactics in Mathlib according to Table 3 and then used LeanDojo to extract data from the modified repository. In this way we were able to generate 41,036 new tactic goal pairs. Since the list of premises in the simplification (simp, dsimp, simpa, simp_rw) tactics is not ordered i.e. the premises can be used in any order to simplify the goal. So, to collect the goals corresponding to each new simplification tactics we used LeanDojo to apply all the new tactics one by one at the goal of the step where the original simplification tactic was supposed to be applied and record the tactic goal pair whichever succeeds and keep on repeating this task on the subgoals generated by the application of first successful tactic and we don't keep the successful tactics in the future list. In this way, we were able to generate 102,682 new unique tactic goal pairs for simplification tactics.

## 4 MODEL

### 4.1 MODEL ARCHITECTURE

We use ByT5-Small as our tactic generator model which has 300m trainable parameters with 12 encoder and 4 decoder layers. ByT5 (Xue et al., 2022) is a tokenizer-free version of Google's T5 (Xue et al., 2020) language model. It operates directly on raw UTF-8 bytes, removing the need for any text preprocessing.

### 4.2 TRAINING OBJECTIVE

The training objective over here is to generate a tactic for the given goal. For example in the theorem 'thm1' discussed in section 3 the input and output for the initial state would be as follows:

```
Input: '''x y z w: nat          Output: "rw [h1]"
(goal)    h1: x = y              (tactic)
          h2: y = z
          h3: z = w
          |- x = w'''
```

For early stopping, we use the validation data set. After each specified number of training steps, we attempt to prove the theorems in the validation dataset using the model being trained as the tactic

predictor and we stop training at the point where we have the highest Pass@1 value on the validation dataset.

## 4.3 PROOF SEARCH

To prove a theorem we use LeanDojo which enables programmatic interaction with Lean. To find the proof of a given theorem we run a proof search. This process begins by selecting the theorem we wish to prove and obtaining its initial goal from Lean. Then, we use the tactic generator model to generate possible tactics that can be applied to the goal. We maintain a proof search tree along with a queue of open goals, sorted by their cumulative log probability as demonstrated in Figure 1. At each step, we expand the goal with the highest cumulative log probability, i.e., the goal for which the tactic generator model is most confident. This method of proof search is called the best-first search.

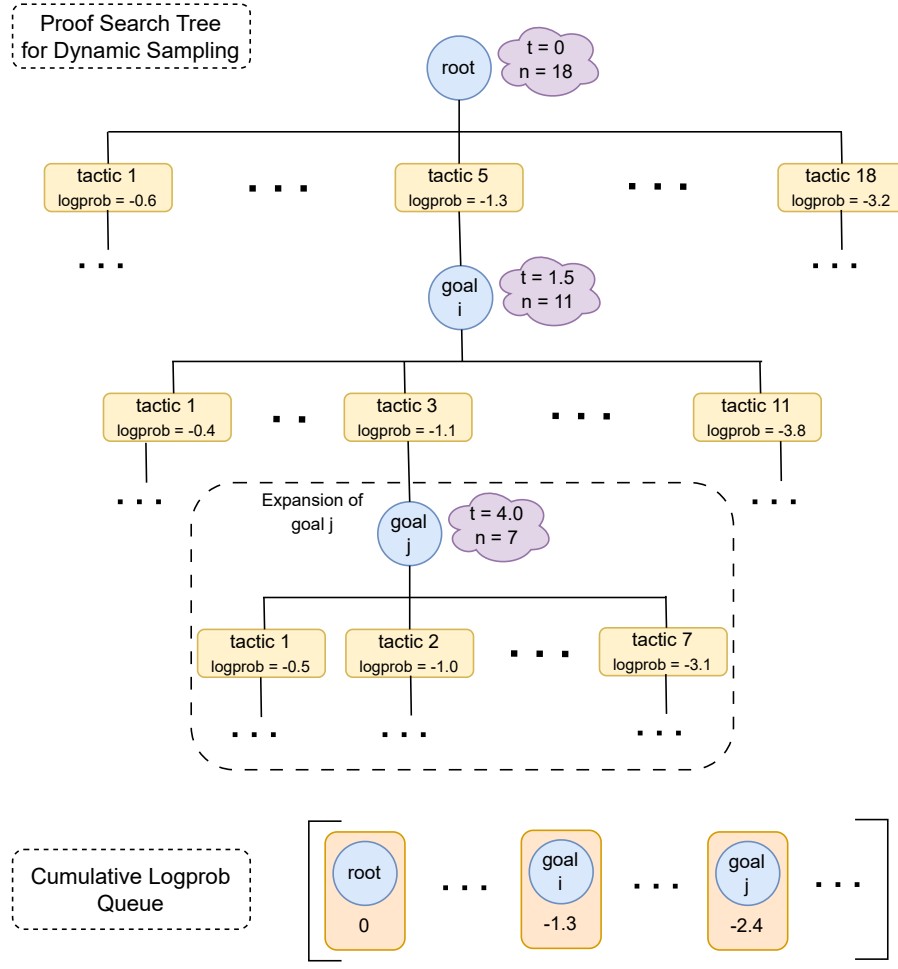

Figure 1: Navigating the Proof Search Tree with Dynamic Sampling: Deciding the number of tactics (n) to sample as time (t) progresses in the proof search, guided by Equation 1.

### 4.3.1 GOAL EXPANSION

Goal expansion is the process of application of tactics to an open goal to generate new subgoals. Unlike the fixed number of samples employed in prior approaches such as (Polu & Sutskever, 2020) and (Yang et al., 2023), we adopt a dynamic sampling approach. In this method, the number of samples is determined dynamically, taking into account the remaining time compared to the total allocated time for theorem proving.

$$n = a + b \cdot e^{-c \cdot r} \qquad \text{Where,} \quad r = \frac{time\_passed}{total\_time} \tag{1}$$

Equation 1 defines the tactic sampling decay function, where $a$, $b$, and $c$ are constants. The constant $a$ determines the minimum number of tactics to sample when the proof search is near the end of the assigned time limit. The constant $b$ determines the initial number of tactics to sample, and the constant $c$ determines the rate of decay in the number of samples. For our experiment we used $a = 6$, $b = 12$ and $c = 5$. The variable $time\_passed$ is the amount of time passed since the start of the proof search, and the variable $total\_time$ is the total amount of time that the proof search is allowed to use (10 minutes in all the cases). The variable $n$ gives the number of tactics to apply to the given goal. Since it is possible that not all of the sampled tactics from the model will be successfully applied to the given goal, we sample $5n$ tactics from the model instead of just $n$. We then apply at most $n$ successful tactics to the goal. In experiments utilizing the fixed sampling method, we sample $n = 64$ tactics at each step and apply all the successful tactics.

Using a dynamic number of tactics in proof search allows us to adjust the balance between exploration and exploitation. At the beginning of the proof search, we start with a higher value of $n$, which means that we are more likely to explore even less promising tactics. As time passes, we reduce $n$ to focus on only the most promising tactics, for which the model has high confidence. This method adds a higher number of nodes to the proof search tree at the beginning, which helps to prevent the search from getting stuck in a state where there are no more nodes to explore (i.e., when the queue of open goals becomes empty). In the end, when we are running out of time to find a proof, this method saves time for both the tactic predictor model and the tactic application process by only asking for a smaller number of the most promising tactics. As time passes, the reduction in the number of tactics to sample also aids in curbing the exponential increase in the number of nodes as the tree depth grows. Consequently, this method can delve deeper into the tree, making it an efficient approach to discovering proofs with a larger number of steps.

## 5 EXPERIMENTS

### 5.1 TRAINING

In our experiments, we used the ByT5-Small model as our tactic generator model, as discussed in Section 4.1. We trained the model using UTF-8 encoding, a batch size of 8, a learning rate of 5e-4, a dropout probability of 0.5, and the AdamW optimizer.

We trained two models, one on the original dataset and the other on an augmented dataset. The original dataset was a collection of tactic-goal pairs extracted from the Mathlib repository without any modification.

For the augmented dataset, we combined three datasets:

1. Tactic-goal pairs extracted from the rewrite modified Mathlib repository.
2. New tactic-goal pairs generated for simplification tactics.
3. Tactic-goal pairs generated from the original Mathlib repository.

We kept the tactic-goal pairs from the original Mathlib repository in the training dataset for the augmented model so that the model could see the original longer tactics (with multiple premises) and learn to use them along with the new smaller tactics (with single premises).

### 5.1.1 EXPERIMENTAL SETUP

We performed all of our experiments on a system with four NVIDIA A100 GPUs, each with 80 GB of VRAM, and 96 CPU cores with 1TB RAM. On this system, collecting tactic-goal pairs from the original and rewrite modified Mathlib repository took around two hours each. Generating tactic-goal pairs for simplification tactics took around two days. Training the tactic generator model on both the original and augmented datasets on the four GPUs took around one day each, and evaluation took around one week.

Table 1: Performance evaluation (Pass@1) of different approaches on test datasets within a 10-minute time constraint across all cases.

| Method | MiniF2F | | ProofNet | | Mathlib | |
|---|---|---|---|---|---|---|
| | Original data | Augmented data | Original data | Augmented data | Original data | Augmented data |
| LeanDojo | 27.8 | - - | 12.8 | - - | - - | - - |
| Optimized LeanDojo | 28.6 | 29.4 | 13.3 | 14.2 | 55.0 | 57.2 |
| DS-Prover + Optimized LeanDojo | 29.0 | **29.8** | 13.7 | **14.2** | 55.0 | **58.8** |

**LeanDojo Optimization:** LeanDojo is designed for both Lean 3 and Lean 4, but to check which version is being used, it used to query the GitHub API, which slowed down the prover. To make it faster, we hardcoded it to always return Lean 3, since we performed all of our experiments in Lean 3. We also increased the tactic application timeout from the default of 5 seconds to 10 seconds for our experiments.

## 5.2 TEST DATASETS

We assess the performance of our methods across a diverse set of test datasets that are designed for testing purposes only and do not contain any theorems that were used in training. This makes them challenging datasets to evaluate on.

**MiniF2F Dataset:** The MiniF2F dataset[2], serves as a comprehensive cross-system benchmark for formal theorem proving within the realm of mathematics. Comprising a total of 489 problem statements, of which 245 constitute the test set and 244 serve as the validation set, this dataset draws its content from esteemed sources such as the American Mathematics Competition (AMC), American Invitational Mathematics Examination (AIME), International Mathematical Olympiad (IMO), as well as materials from high-school and undergraduate mathematics courses.

**ProofNet Dataset:** ProofNet[3] is a benchmark dataset for formally proving theorems in undergraduate-level mathematics. It consists of 374 examples, each of which is a formal theorem statement in the Lean 3 theorem prover. The problems are drawn from popular undergraduate pure mathematics textbooks and cover topics such as real and complex analysis, linear algebra, abstract algebra, and topology.

**Mathlib Dataset:** We also tested our methods on the Mathlib test dataset, which is a dataset we created by random selection of theorems from the Mathlib repository, as elaborated in Section 3.2. It is important to highlight that, due to constraints in our computing resources, we limited our evaluation to a subset of this dataset, comprising 500 theorems, from the original set of 1000 test theorems.

## 5.3 RESULTS

Table 1 summarizes the performance evaluation (Pass@1) of both models on all of our test datasets using various methods. Table 2 compares the performance of our approach to other baselines on the MiniF2F test dataset. We could not reproduce previous works due to the high computational requirements of most of them, so we used the performance reported in their papers for our comparison. As depicted in the table, we have the best performance of 29.8% on **MiniF2F**, which slightly surpasses the best-reported Pass@1 of 29.6% (Polu et al., 2022) using Lean.

On the **ProofNet** test dataset, we have the best performance (Pass@1) of **14.2%** which is higher than the only reported performance of 13.8% with ReProver (Yang et al., 2023).

## 5.4 DISCUSSION

Table 1 indicates that using the model trained on augmented data consistently outperforms the model trained solely on the original data. The augmented model's training with a larger number of tactic-

---

[2]https://github.com/openai/miniF2F/tree/main/lean
[3]https://github.com/zhangir-azerbayev/ProofNet

Table 2: Comparison with Baseline Performance (Pass@1) on the Testing Data of Lean MiniF2F.

| S. No. | Method | Pass@1 |
|---|---|---|
| 1 | Lean GPT-f (Zheng et al., 2021) | 24.6 |
| 2 | PACT (Han et al., 2021) | 24.6 |
| 3 | Lean Expert Iteration (Polu et al., 2022) | 29.6 |
| 4 | ReProver (Yang et al., 2023) | 26.5 |
| 5 | Augmented data + DS-Prover + Optimized LeanDojo (Ours) | **29.8** |

goal pairs provides it with the capability to effectively handle diverse goals. This underscores the augmented dataset's effectiveness in training the tactic generator model for Neural Theorem Proving.

We can also observe from the table that employing dynamic sampling, instead of fixed sampling, consistently enhances performance in most cases for both the augmented and original datasets. The dynamic sampling method either outperforms the fixed sampling or performs equally well, but it never exhibits worse performance. This aspect makes the use of the dynamic sampling method a promising new approach for neural theorem proving.

To study the impact of employing dynamic sampling and to ascertain its enhancements in performance, we conducted an analysis of the theorems' proofs and the search trees generated for all the theorems within our mathlib test dataset. Specifically, we aimed to comprehend the disparity between the tree structures produced by both dynamic and fixed sampling methods for proof searching. A comparative plot (Figure 2) depicting the average number of nodes at each level (tree depth) for both methods was generated for this purpose. Upon examining the plot, noticeable differences emerged: the fixed sampling method exhibited a rapid increase in the number of nodes initially, reaching its peak at a depth of 3. In contrast, the dynamic sampling method showcased a slower increase, reaching its peak at depth 4. Overall, the plot suggests that the fixed sampling method tends to prioritize shorter proofs, whereas the dynamic sampling method displays a more extensive distribution. It actively explores varied proof lengths, aiding in the discovery of even longer proofs.

For the theorems successfully proven within the allocated time frame, we generated plot 3, which compares proof sizes between those achieved using dynamic sampling and fixed sampling, juxtaposed with the original proof sizes for reference. Our analysis revealed that both methods demonstrated comparable performance for smaller-sized proofs (consisting of 1 or 2 steps). However, the ratio of the number of theorems proved using dynamic and fixed sampling methods increases as the proof size increases. This observation indicates that the dynamic sampling method optimizes time utilization by delving into deeper levels of the proof search tree. Consequently, it efficiently explores and discovers longer proofs, a capability not as effectively realized by the fixed sampling method.

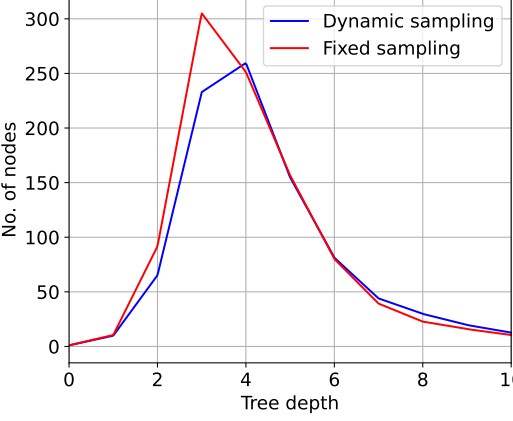

Figure 2: Number of nodes in the search tree at various depths

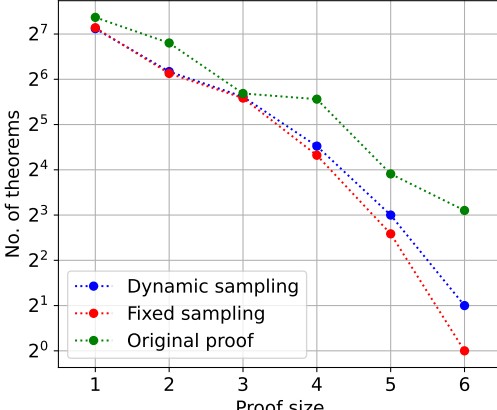

Figure 3: Comparison of proof sizes for theorems proven using dynamic and fixed sampling methods

Our website functions as a convenient online platform allowing users to input their Lean problem statement, after which the prover will make an attempt to provide a proof within the allocated time frame. For a detailed exploration of how our website facilitates theorem proving, please consult Section A.2 in the Appendix.

## 6 CONCLUSION

We have introduced DS-Prover, a dynamic sampling method for Neural Theorem Proving. This method dynamically determines the number of tactics to apply to expand the current goal, taking into account the remaining time compared to the total allocated time for proving a theorem. We also studied the effect of augmenting the training dataset by decomposing tactics with multiple premises into tactics with single premises. Our experiments show that using the model trained on the augmented dataset has better performance than using the model trained only on the original dataset. This establishes our neural theorem proving methods as a new and promising approach. Further, we release a public website for theorem proving in Lean 3.

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

# A APPENDIX

## A.1 DATA AUGMENTATION

| Original tactic | Replaced tactics |
|---|---|
| rw $[p_1,...,p_n]$ | rw $[p_1]$; ...; rw $[p_n]$ |
| rw $[p_1,...,p_n]$ at $h_1$ | rw $[p_1]$ at $h_1$; ...; rw $[p_n]$ at $h_1$ |
| erw $[p_1,...,p_n]$ | erw $[p_1]$; ...; erw $[p_n]$ |
| erw $[p_1,...,p_n]$ at $h_1$ | erw $[p_1]$ at $h_1$; ...; erw $[p_n]$ at $h_1$ |
| rwa $[p_1,...,p_n]$ | rwa $[p_1]$; ...; rwa $[p_n]$ |
| rwa $[p_1,...,p_n]$ at $h_1$ | rwa $[p_1]$ at $h_1$; ...; rwa $[p_n]$ at $h_1$ |
| equiv_rw $[p_1,\ p_2]$ | equiv_rw $[p_1]$; ...; equiv_rw $[p_2]$ |
| equiv_rw $[p_1,...,p_n]$ at $h_1$ | equiv_rw $[p_1]$ at $h_1$; ...; equiv_rw $[p_n]$ at $h_1$ |
| assoc_rw $[p_1,...,p_n]$ | assoc_rw $[p_1]$; ...; assoc_rw $[p_n]$ |
| assoc_rw $[p_1,...,p_n]$ at $h_1$ | assoc_rw $[p_1]$ at $h_1$; ...; assoc_rw $[p_n]$ at $h_1$ |
| simp_rw $[p_1,...,p_n]$ | simp_rw $[p_1]$; ...; simp_rw $[p_n]$ |
| simp_rw $[p_1,...,p_n]$ at $h_1$ | simp_rw $[p_1]$ at $h_1$; ...; simp_rw $[p_n]$ at $h_1$ |
| simp $[p_1,...,p_n]$ | simp $[p_1]$; ...; simp $[p_n]$ |
| simp $[p_1,...,p_n]$ at $h_1$ | simp $[p_1]$ at $h_1$; ...; simp $[p_n]$ at $h_1$ |
| simp only $[p_1,...,p_n]$ | simp only $[p_1]$; ...; simp only $[p_n]$ |
| simp only $[p_1,...,p_n]$ at $h_1$ | simp only $[p_1]$ at $h_1$; ... |
| dsimp $[p_1,...,p_n]$ | dsimp $[p_1]$; ...; dsimp $[p_n]$ |
| dsimp $[p_1,...,p_n]$ at $h_1$ | dsimp $[p_1]$ at $h_1$; ...; dsimp $[p_n]$ at $h_1$ |
| dsimp only $[p_1,...,p_n]$ | dsimp only $[p_1]$; ...; dsimp only $[p_n]$ |
| dsimp only $[p_1,...,p_n]$ at $h_1$ | dsimp only $[p_1]$ at $h_1$; ... |
| simpa $[p_1,...,p_n]$ | simpa $[p_1]$; ...; simpa $[p_n]$ |
| simpa $[p_1,...,p_n]$ at $h_1$ | simpa $[p_1]$ at $h_1$; ...; simpa $[p_n]$ at $h_1$ |
| simpa only $[p_1,...,p_n]$ | simpa only $[p_1]$; ...; simpa only $[p_n]$ |
| simpa only $[p_1,...,p_n]$ at $h_1$ | simpa only $[p_1]$ at $h_1$; ... |

Table 3: Description of new tactic generation

## A.2 WEBSITE DESCRIPTION

| S.No. | Description | Figure |
|---|---|---|
| 1 | Home Page | Figure 4 |
| 2 | Theorem Proving in Action | Figure 5 |
| 3 | Found a Proof | Figure 6 |
| 4 | Another Example of Proving a Theorem | Figure 7 |
| 5 | Proof Search Timed Out | Figure 8 |
| 6 | Error Report | Figure 9 |
| 7 | Setting up Local Environment | Figure 10 |

Table 4: Table Summarizing the Usage of Website for Theorem Proving.

# DS-Prover

A Dynamic Sampling Based Neural Theorem Prover

**Your statement in Lean 3:**

```
variables x y z w : nat

theorem thm1 (h1 : x = y) (h2 : y = z) (h3 : z = w) : x = w :=
```

Prove

Figure 4: This is the home page of the theorem prover, featuring a text box where we can input our formal statement in Lean 3. The website's backend will then attempt to prove it within the specified time limit.

**Your statement in Lean 3:**

```
variables x y z w : nat

theorem thm1 (h1 : x = y) (h2 : y = z) (h3 : z = w) : x = w :=
```

Prove

**Proving theorem**

10 / 300 seconds

Figure 5: The website is attempting to prove the given theorem.

**Your statement in Lean 3:**

```
variables x y z w : nat

theorem thm1 (h1 : x = y) (h2 : y = z) (h3 : z = w) : x = w :=
```

Prove

## Found a Proof!

**Statement:**
import all
variables x y z w : nat

theorem thm1 (h1 : x = y) (h2 : y = z) (h3 : z = w) : x = w :=

**Proof:**
begin
    rw [h1, h2, h3],
end

**Time:** 3.9 seconds

Figure 6: A proof returned by the website when it finds one.

# DS-Prover

A Dynamic Sampling Based Neural Theorem Prover

**Your statement in Lean 3:**

```
def even_fun (f : ℝ → ℝ) := ∀ x, f (-x) = f x
def odd_fun (f : ℝ → ℝ) := ∀ x, f (-x) = -f x

example (f g : ℝ → ℝ) : even_fun f → even_fun g →  even_fun (f + g) :=
```

Prove

## Found a Proof!

**Statement:**
import all
def even_fun (f : ℝ → ℝ) := ∀ x, f (-x) = f x
def odd_fun (f : ℝ → ℝ) := ∀ x, f (-x) = -f x

example (f g : ℝ → ℝ) : even_fun f → even_fun g → even_fun (f + g) :=

**Proof:**
begin
    unfold even_fun,
    intros h x,
    simp [h, x],
end

**Time:** 56 seconds

Figure 7: Illustration of the Website Successfully Proving Another Theorem.

**Your statement in Lean 3:**

```
theorem mathd_algebra_209
  (σ : equiv ℝ ℝ)
  (h₀ : σ.2 2 = 10)
  (h₁ : σ.2 10 = 1)
  (h₂ : σ.2 1 = 2) :
  σ.1 (σ.1 10) = 1 :=
```

**Prove**

## Search Timed Out!

Could not find a proof in the given time limit.

**Statement:**
theorem mathd_algebra_209
(σ : equiv ℝ ℝ)
(h₀ : σ.2 2 = 10)
(h₁ : σ.2 10 = 1)
(h₂ : σ.2 1 = 2) :
σ.1 (σ.1 10) = 1 :=

**Proof:**
None

**Time:** 329.6 seconds

Figure 8: The prover may not always find a proof for the given statement within the provided time limit; this is one such example.

**Your statement in Lean 3:**

```
theorem exercise_13_6 :
  ¬ (∀ U, Rl.is_open U → K_topology.is_open U) ∧ ¬ (∀ U, K_topology.is_open U →
Rl.is_open U) :=
```

**Prove**

## Encountered an Error!

**Error:** unknown identifier 'Rl.is_open'

Figure 9: If there is any issue with the provided statement, the prover returns an error along with a description. In this specific example, the identifier 'Rl.is_open' is not defined.

**Your statement in Lean 3:**

```
def lower_limit_topology (X : Type) [preorder X] :=
  topological_space.generate_from {S : set X | ∃ a b, a < b ∧ S = Ico a b}

def Rl := lower_limit_topology ℝ

def K : set ℝ := {r | ∃ n : ℕ, r = 1 / n}

def K_topology := topological_space.generate_from
  ({S : set ℝ | ∃ a b, a < b ∧ S = Ioo a b} ∪ {S : set ℝ | ∃ a b, a < b ∧ S = Ioo a b \ K})

theorem exercise_13_6 :
  ¬ (∀ U, Rl.is_open U → K_topology.is_open U) ∧ ¬ (∀ U, K_topology.is_open U →
  Rl.is_open U) :=
```

Prove

**Proving theorem**

70 / 300 seconds

Figure 10: We can set up our local environment of definitions and variables before formulating the theorem statement.

