# OpenReview forum: "DS-Prover: A Dynamic Sampling Based Approach for Neural Theorem Proving"
_ICLR.cc/2024/Conference — Submitted to ICLR 2024_

### Official Review · Reviewer_MSm4 · 2023-10-14

**Soundness:** 1 poor
**Presentation:** 2 fair
**Contribution:** 2 fair
**Rating:** 3
**Confidence:** 5

**Summary:**

The paper fine-tunes a language model on Lean3 state-tactic pairs and uses it to search for proofs on college-to-university-level mathematical problems. The main innovations are an augmentation of the training data and an inference-time technique that dynamically chooses how many branches to expand depending on the remaining time.

**Strengths:**

The proposed methods are straightforward and easy to implement. They bring performance improvement on the miniF2F benchmark.

**Weaknesses:**

I'm not entirely convinced by the claims that these changes are truly beneficial, or how much improvement to the baselines there is:
- Data augmentation: In table 1, one can actually see that the data augmentation helps for problems in miniF2F, but is actually detrimental to performance for problems in ProofNet (which are harder) and problems in Mathlib (which cover wider mathematical domains). Therefore, the conclusion ought to be that the data augmentation might improve things for certain problems, but cannot increase performance across the board or in general.
- Without the data augmentation, which I think should be considered as a non-general method as the above point suggests, the proposed method proves 29.0\% of problems on miniF2F test, which is lower than Lean + Expert Iteration by Polu et al. (2022).

I have more detailed comments and suggestions below:

1. > End of page 1. "To overcome the issue of limited data for training the model, various alternative attempts have been explored to improve the performance of automated theorem provers, such as reinforcement learning Polu & Sutskever (2020)"

    (Minor) It is slightly strange to refer to it as an alternative attempt since Polu & Sutskever (2020) is arguably the first work to use generative transformer with interactive theorem provers.

2. (Major) One of the major contributions is "we also release a public theorem prover website", but this website is not provided and therefore it is impossible to assess this claim.

3. (Minor) Scholarship needs improving: the related works should not be a simple stack of papers covering related topics, but should rather compare and contrast other works with this current work.

4. (Minor) Citation style: A lot of instances where the citation is glued to the text with no separation, e.g., Leande Moura et al. (2015) on page 3. Use ~\citep to cite the paper and ~\citet to cite the authors.

5. > Page 4 assumption: The assumption tactic is used to prove the goal by assuming it’s true based on the available hypotheses

    (Minor) This is clearly not accurate. One never assumes the goal to be true. Rather, one matches the goal with the assumptions with this tactic.

7. (Critical) Misleading claim: I'm surprised that the authors mentioned the HTPS paper by Lample et al., but not their results. The HTPS paper achieved a success rate of >40\% on the miniF2F with pass@64, compared to 30\% in this paper with pass@1. Of course, the success rates between pass@1 and pass@64 are very different, but one should be very careful before making the claim **We achieve a new state-of-the-art performance of 30.6% on MiniF2F using Lean** if its success rate is 10\% (absolute), or 25\% (relative) lower than a paper published 17 months ago. Some experiments for DS-solver at a higher pass@k should be performed before such claims can be verified.

**Questions:**

For how many steps was the model trained for? What are the training and validation metrics? What is the experimental wallclock time limit per problem for miniF2F, ProofNet and Mathlib?

---

> ### Author Response · Authors · 2023-11-18
> **Response to reviewer MSm4**
>
> We thank the reviewer for their feedback and will respond to the raised questions below.
>
> **Weakness:**
>
> **W1:** Thanks for pointing this out, we now have improved our statement.
>
> **W2:** We haven't been able to find a secure way to host and share the website without disclosing our identity. As a result, we have refrained from sharing the website at this time.
>
> **W3:** Thanks for the suggestion we are are improving the related works section and also will be adding the related works of theorem provers other than Lean in our paper.
>
> **W4:** Thank you for pointing this out, we now have corrected the citations in the paper.
>
> **W5:** Thank you for highlighting this mistake, we now have corrected our statement in the paper.
>
> **W6:** Thank you for pointing this out. We acknowledge that our best performance is only for pass@1, as we did not conduct experiments for higher pass@k. We will correct our statement accordingly.
>
>
> **Questions:**
>
> **Q1:** For early stopping, we used the Pass@1 of the validation dataset. After each 0.5 epoch in the augmented data model (1 epoch in the original data model) we used the current model to attempt to prove the theorems in the validation dataset. And we selected the model with the best Pass@1 on the validation data for our final evaluation.
>
> We had given a time limit of 10 minutes to find proof for all the theorems in each dataset.
>
> **We will be submitting our updated paper soon. Thanks for your patience.**

---

> > ### Comment · Reviewer_MSm4 · 2023-11-19
> >
> > Thank you for your reply. I shall be waiting for the updated paper before considering adjusting my rating.

---

### Official Review · Reviewer_6Nhd · 2023-10-28

**Soundness:** 2 fair
**Presentation:** 2 fair
**Contribution:** 2 fair
**Rating:** 3
**Confidence:** 3

**Summary:**

Aiming to generate tactics for interactive theorem provers (ITPs) with the help of large language models (LLMs), the paper proposes two methods: the dynamic sampling that determines how many tactics are investigated depending on the remaining time, and the data augmentation that splits an application of a specific tactic with multiple arguments into multiple applications of the tactic with one argument. The effect of the proposed methods is experimentally shown on certain datasets of mathematical theorem proving.

**Strengths:**

- The paper provides several examples of proof code and tactics in Lean. It would help readers unfamiliar with Lean find the aim and method studied in the paper.
- The experiments show that the proposed approaches can improve the proof search with LLMs.

**Weaknesses:**

- The improvement by the dynamic sampling seems incremental. The "Original data" columns in Table 1 shows only the improvement of 0.4 points against the optimized LeanDojo.
- The effect of the data augmentation is not entirely clear because there is no experiment that employs only it.
- Not all the experimental settings are clear. Specifically, I cannot find how many tactics are sampled in the fixed sampling.
- I'm not convinced by the discussion for Figure 2. It shows that the difference between the dynamic and fixed sampling methods are almost fixed. I suspect it means that the dynamic sampling is effective only for theorems with short proofs because, even when the time budgets are increased, the difference is retained (rather, becomes small).
- The paper cites other works without parenthesizing the author names. It lowers the readability.

**Questions:**

- How many tactics are sampled in the fixed sampling? Does changing the number of sample tactics influence the result?
- Does Figure 2 mean that the dynamic sampling is effective only for theorems with short proofs?

---

> ### Author Response · Authors · 2023-11-18
> **Response to reviewer 6Nhd**
>
> We thank the reviewer for the helpful feedback and will address the raised concerns below.
>
> **Weakness:**
>
> **W1:** Although the increment in performance may be smaller in some cases, we can also observe that dynamic sampling consistently either outperforms fixed sampling or performs equally well, never exhibiting inferior performance. This consistency makes dynamic sampling a better choice of algorithm for searching proofs within a given time limit.
>
> **W2:** Using only the augmented data would result in longer proofs since all the generated tactics will only have a maximum of one premise. Consequently, we would need to search the tree in greater depth to find proof, which would increase the time required for the search. Therefore, we decided not to perform this experiment.
>
> **W3:** Thanks for pointing this out. In the fixed sampling method, we sampled 64 tactics each time. We have now included this information in the paper.
>
> **W4:** We have conducted experiments to study this matter in detail, and we will be discussing the findings in the paper.
>
> **W5:** Thank you for pointing this out, we now have corrected the citations in the paper.
>
> **Questions:**
>
> **Q1:** In fixed sampling, 64 tactics are sampled each time.
>
> Yes, decreasing the number of tactics to sample and apply as time passes influences the result. We have added new results discussing this aspect in the paper.
>
> **Q2:** No, in fact, our analysis of the results demonstrates that the reverse is true. We have now included these results in the paper.
>
> **We will be submitting our updated paper soon. Thanks for your patience.**

---

> > ### Comment · Reviewer_6Nhd · 2023-11-23
> >
> > I thank the authors for giving the response and updating the paper. I looked at the updated paper, but I don't think it is at the stage to be published due to the following concerns.
> >
> > - The updated paper shows the result in converting tactics to standard forms. Therefore, it is not clear that the improvement of the performance in the updated paper is owing to the data augment, the standardization, or both of them. Especially, in the original paper, the data augmentation does not contribute to the improvement of the performance on ProofNet and Mathlib, while in the updated paper, it does. Given only this result, I cannot ignore the possibility that the standardization is more important than the data augmentation.
> >
> > - The updated paper claims that the dynamic sampling is effective especially on longer proofs (Figure 3). However, I'm unsure how it can be made consistent with Figure 2 in the original paper which says that the dynamic sampling can solve more problems than the fixed sampling even in short time (2.5 minutes).
> >
> > - I'm not very convinced by the response to W2. Do the authors mean using only the data augmentation is definitely useless?

---

> > > ### Author Response · Authors · 2023-11-23
> > > **Answer to the concerns raised by reviewer 6Nhd**
> > >
> > > We would like to thank the reviewer for raising their concerns. We will address them below:
> > >
> > > **Dynamic and Fixed Sampling:**
> > >
> > > To clarify, Figure 2 in the original paper demonstrates that, across various specified time limits, dynamic sampling outperforms fixed sampling. Meanwhile, Figure 3 in the updated paper reveals that, within a given time limit, our analysis discovered comparable performance between both methods for smaller-sized proofs. However, the ratio of theorems proven using dynamic and fixed sampling methods increases as the proof size grows. This observation suggests that the dynamic sampling method optimizes time utilization by exploring deeper levels of the proof search tree.
> > >
> > > Combining these findings, we can conclude that, within any given time period, the dynamic sampling method can delve deeper into the tree compared to fixed sampling. Consequently, it can uncover longer proofs while also efficiently identifying shorter ones. In contrast, fixed sampling primarily focuses on shorter proofs.
> > >
> > > It's important to note that proof size becomes a relative term concerning time. For instance, a proof of size 3 might seem lengthy within a 2.5-minute time limit, whereas it might be considered an average size (not significantly longer) within a 10-minute limit. This difference arises because there is more time available to explore the tree, allowing both algorithms to easily reach a depth of 3 within 10 minutes. This is a challenge within the 2.5-minute time constraint.
> > >
> > > We hope this explanation addresses your concerns regarding the sampling methods.

---

### Official Review · Reviewer_FKE8 · 2023-10-31

**Soundness:** 3 good
**Presentation:** 2 fair
**Contribution:** 2 fair
**Rating:** 6
**Confidence:** 4

**Summary:**

In this submission, the authors consider automatic theorem proving with Transformer models. They introduce a dynamic way of sampling from a tactic space while take total time left for proving the theorem into account. They show that this makes proof search more efficient by balancing exploration vs exploitation. They additionally provide a data augmentation by decomposing tactics with multiple premises.
They conducted experiments by training a ByT5 model on formalized theorems in Lean (mathlib repo) and evaluating their model on MiniF2F and ProofNet, two standard datasets in the literature. The results show that they approach is resulting in performance gains.

**Strengths:**

- Interactive theorem proving (especially in Lean) has gained a lot of attention recently. With more and more mathematicians picking it up and more and more machine learning support to ease the construction of the proofs, the paper is certainly relevant for ICLR and the problem is interesting.
- Although the models are fairly small, they provide a new state of the art
- The approach is straightforward and well-explained. Datasets are open-source; and they seemed to released their models on a public website (this makes the experiments reproducible for academics and students)

**Weaknesses:**

- The comparisons in Table 1 seem to be slightly unfair: (time used by LeanDojo and the optimized version vs. the proposed method is unclear)
- Not clear from the paper, if model weights and code will be open-sourced
- The contribution is straightforward and more on the minor side
- Interesting ablations and more in-depth analysis is missing (more detailed analysis of time tradeoff in Figure 2)
- The related work is highly insufficiently discussed

**Questions:**

- Will the code and models be open-source?
- Would the authors consider expanding the related work section a bit including highly influential, but more non-lean related research?
- what is the time used by LeanDojo and the optimized version vs. the sampling? Can this be added to Table 1
- Where are the tradeoffs in exploration vs exploitation; Table 2? For example, what size are the proofs where the sampling strategy works well? Are there cases where fixed sampling is better?

---

> ### Author Response · Authors · 2023-11-18
> **Response to reviewer FKE8**
>
> We thank the reviewer for their feedback and will respond to the raised questions below.
>
> **Weakness:**
>
> **W1:** We apologize for the confusion. To clarify, we maintained a consistent time limit of 10 minutes for all the experiments conducted.  This information has now been included in the table caption for clarification.
>
> **W2:** Yes, we plan to make the Model, Code, and DataSets available as open-source.
>
> **W3:** We acknowledge that the contributions may seem straightforward. However, by implementing these small improvements, we've achieved comparable performance to models with significantly higher computational requirements. This aspect makes our method highly suitable for general use, even with limited computational resources.
>
> **W4:** We've conducted additional experiments within this context and intend to incorporate these findings into our paper.
>
> **W5:** Thank you for bringing up this point. We're enhancing our related works section by including further discussions on papers related to theorem provers beyond Lean.
>
>
> **Questions:**
>
> **Q1:** Yes, we plan to make the Model, Code, and DataSets available as open-source.
>
> **Q2:** Yes, we're enhancing our related works section by including further discussions on papers related to theorem provers beyond Lean.
>
> **Q3:** In every case, we set a time limit of 10 minutes.
>
> **Q4:** **1, 2.** We now have added some more plots discussing these aspects in the paper.
>
> **3.** In our experiments, dynamic sampling consistently either outperforms fixed sampling or performs equally well, never exhibiting inferior performance.
>
> **We will be submitting our updated paper soon. Thanks for your patience.**

---

### Official Review · Reviewer_X3v4 · 2023-11-01

**Soundness:** 3 good
**Presentation:** 2 fair
**Contribution:** 2 fair
**Rating:** 3
**Confidence:** 4

**Summary:**

This paper introduces DS-Prover, a automated theorem proving framework in the Lean proof assistant. The main feature of this framework is that it dynamically determines the number of tactics to explore taking into account the remaining time resources. Performance gain has been demonstrated against the previous Fixed Sampling strategy in the LeanDojo paper.

**Strengths:**

The idea of taking time into the explore-exploit tradeoff is novel and of practical significance. I especially appreciate the authors' effort of pushing the boundary of low-budget neural theorem proving, which could be much more useful to daily ITP users.

**Weaknesses:**

- The writing can still be improved. For example, the \cite or \citep should not be used interchangeably, and there should be a space before each inline citation. Also, some sentences can use some polishing, e.g., 'where anyone can put the formal statement in Lean for their mathematical theorem' on page 2.
- Related prior work in tactic prediction in other systems (Coq, HOL4) should have been mentioned and compared. In particular, some considerations between atomic and compound tactics have been discussed in prior work in Coq (https://arxiv.org/abs/1905.09381, https://proverbot9001.ucsd.edu).
- The claim of 'a new state-of-the-art performance of 30.6% on MiniF2F using Lean' is not entirely accurate, as the HyperTree Proof Search (HTPS) paper has already achieved over 40% success rate over the same dateset. I understand that your approach does not use reinforcement learning nor the same amount of computation resources as in the HTPS paper, but it might be better to make those assumptions clear and perhaps draw a more detailed comparison against HTPS.

**Questions:**

- Table 1: one of the contributions of LeanDojo was to propose the novel_premises benchmark, which is believed to better reflect the generalization ability of the proof agent. Is is possible to have DS-Prover also run on it?
- Discussions: would it be possible to have a length distribution comparisons between the generated proofs from Dynamic Sampling and Fixed Sampling models? Some qualitative examples to illustrate the differences between these two sampling methods would be highly appreciated.

---

> ### Author Response · Authors · 2023-11-18
> **Response to reviewer X3v4**
>
> We thank the reviewer for the helpful feedback and will address the raised concerns below.
>
> **Weakness:**
>
> **W1**: Thanks for pointing this out we now have corrected the citations and also made other suggested changes in the paper.
>
> **W2:** Thanks for the suggestion, we will be adding the related works of theorem provers other than Lean in our paper.
>
> **W3:** Thank you for pointing this out. We acknowledge that our best performance is only for pass@1, as we did not conduct experiments for higher pass@k. We will correct our statement accordingly.
>
>
> **Questions:**
>
> **Q1:** Yes, we can run DS-Prover on the novel_premises benchmark. However, we wanted to assess the performance of our methods on a general dataset, not on a specifically prepared dataset that uses premises unseen by the model in its proof, hence we didn't consider running it on the novel_premises benchmark.
>
> **Q2:** Yes, we have compared the lengths of proofs in Dynamic Sampling and Fixed Sampling methods, and we will be including these comparisons in the paper.
>
> **We will be submitting our updated paper soon. Thanks for your patience.**

---

> > ### Author Response · Authors · 2023-11-21
> >
> > **[Weakness]** We have updated our paper to incorporate the improvements suggested for weaknesses W1, W2, and W3.
> >
> > **[Q2]**
> > We have included two plots in the discussion section that compare the tree structure and proof sizes between the dynamic and fixed sampling methods. Our findings from these plots indicate that the number of nodes at various depths of the tree is more widely distributed in dynamic sampling. This broader distribution allows for exploration of proofs of varying lengths, unlike the fixed sampling method which primarily focuses on shorter proofs.
> >
> > Another plot, comparing the proof sizes of theorems proven using both methods, reveals that the ratio of proof sizes between dynamic and fixed sampling increases as the size of the proofs becomes longer. This observation suggests that dynamic sampling excels at exploring even longer proofs compared to fixed sampling.
> >
> > **We have submitted the revised paper and kindly request the reviewer to assess our updates and adjust the ratings accordingly.**

---

> > > ### Comment · Reviewer_X3v4 · 2023-11-21
> > > **Thank you for the responses**
> > >
> > > I thank the authors' effort in responding to my queries as well as revising the paper. The two extra plots make sense -- I super appreciate them. Nevertheless, I still don't think the paper is of publishable quality at this stage -- more experiments could be used to demonstrate the effectiveness of dynamic sampling. Experiments of dynamic sampling on other tactic-style systems like Coq and HOL4 might be beneficial.

---

### Author Response · Authors · 2023-11-21
**Updated Experiments and Results in Our Paper**

We would like to emphasize to the reviewer that we repeated our experiments on the augmented dataset. Initially, our model was trained without converting the tactics of the augmented data into their standard form. Subsequently, we noticed an issue wherein during the sampling process of **n** number of tactics from the model, we frequently encountered repeated tactics in both their standard and non-standard forms (e.g., 'rw [h1]' and 'rw [  h1]'). While these tactics appeared different for the model, they were identical for the Lean prover.

Upon recognizing this discrepancy, we repeated our experiment for the augmented dataset. This time, we converted all the tactics into their standard form before training the model. Subsequently, we used this new trained model to evaluate all the test datasets. **Our paper has been updated with the new results.**

**We kindly request the reviewers to review our updated paper and adjust the ratings accordingly.**

---

### Meta-Review · Area_Chair_NvL5 · 2023-12-12

**Metareview:**

This work introduces an automated theorem prover in Lean, that dynamically determines the number of tactics to explore taking into account the remaining time resources. Reviewers think the idea is novel and easy to implement, but the related works, evaluation, and in-depth analysis are far from sufficent to demonstrate its effectiveness. AC agrees in its current version this work is not ready to publish.

**Justification For Why Not Higher Score:**

The related works, evaluation, and in-depth analysis are far from sufficent to demonstrate its effectiveness.

**Justification For Why Not Lower Score:**

N/A

---

### Decision · Program_Chairs · 2024-01-16

Reject